# Impact of Low-Reactivity Calcined Clay on the Performance of Fly Ash-Based Geopolymer Mortar

Kwabena Boakye and Morteza Khorami *

School of Energy, Construction & Environment, Faculty of Engineering & Computing, Coventry University, Coventry CV1 5FB, UK; boakyek4@uni.coventry.ac.uk
* Correspondence: morteza.khorami@coventry.ac.uk

**Abstract:** Availability of aluminosiliceous materials is essential for the production and promotion of geopolymer concrete. Unlike fly ash, which can only be found in industrial regions, clays are available almost everywhere but have not received sufficient attention to their potential use as a precursor for geopolymer synthesis. This study investigates the effectiveness of calcined clay as a sole and binary precursor (with fly ash) for the preparation of geopolymer mortar. Fly ash-based geopolymer containing between 0 and 100% low-grade calcined clay was prepared to investigate the effect of calcined clay replacement on the geopolymerization process and resultant mortar, using a constant liquid/solid ratio. Reagent-grade sodium hydroxide (NaOH) and sodium silicate ($Na_2SiO_3$) were mixed and used for the alkali solution preparation. Six different mortar mixes were formulated using sand and the geopolymer binder, comprising varying fly ash-to-calcined clay ratios. The combined effect of the two source materials on compressive strength, setting time, autogenous shrinkage, and porosity was studied. The source materials were characterized using XRD, SEM, FTIR, and XRF techniques. Isothermal calorimetry was used to characterize the effect of low-grade calcined clay on the geopolymerization process. The addition of calcined clay reduced the surface interaction between the dissolved particles in the alkali solution, leading to slow initial reactivity. Geopolymer mortar containing 20% calcined clay outperformed the reference geopolymer mortar by 5.6%, 17%, and 18.5% at 7, 28, and 91 days, respectively. The MIP analysis revealed that refinement of the pore structure of geopolymer specimens containing calcined clay resulted in the release of tensional forces within the pore fluid. Optimum replacement was found to be 20%. From this study, the mutual reliance on the physical and inherent properties of the two precursors to produce geopolymer mortar with desirable properties has been shown. The findings strongly suggest that clay containing low content of kaolinite can be calcined and added to fly ash, together with appropriate alkali activators, to produce a suitable geopolymer binder for construction applications.

**Keywords:** calcined clay; geopolymer; mortar; fly ash; shrinkage; strength; porosity

## 1. Introduction

The construction industry, being a major contributor of $CO_2$ to the atmosphere through the production of Portland cement, has been under immense pressure to come up with materials and processes that are eco-friendly and promote sustainable development. In response, several steps have been taken to reduce carbon emissions and depletion of natural resources, including (i) having alternate use for industrial and agricultural waste materials through recycling and (ii) promotion and utilization of eco-friendly materials for housing and infrastructure [1]. In this regard, chemically activated materials (geopolymers) have gained massive recognition over the years due to their technical, and to a large extent, environmental benefits [2]. Geopolymer binders are produced from chemical reactions between aluminosilicate-rich precursors and alkali solutions at room or high temperatures [3–5].

Alkali-activated cements originally appeared more than five decades ago, as evidenced by tests conducted by Purden and Glukhovsky [6]. Davidovits subsequently developed and advanced this in the form of geopolymer, employing low calcium alkali-activated binders [7]. Several studies have shown superior mechanical and durability characteristics of alkali-activated geopolymer concrete to plain concrete [8–10]. However, one drawback of utilizing alkali-activated materials is that it requires heat curing to achieve strength. Some researchers have, in recent times, cured at room temperature and yet have recorded significant improvement in compressive strength [11,12].

Geopolymer can be made from a wide range of aluminosilicate materials, although fly ash, slag, and metakaolin have been widely researched and utilized among them [13–16]. However, recently, fly ash has become scarce as a result of the closing of several coal-fired power generation plants around the world. A complete shutdown of these power stations is imminent and could potentially lead to global shortages, making fly ash unavailable for applications in the construction industry [17,18]. Another material that has produced excellent geopolymer concrete properties is metakaolin, prepared from the calcination of clays with high kaolinite content [19–23]. However, the challenge with kaolinitic clay, also referred to as China clay, is that it can only be found in specific places and is quite expensive due to its use in the ceramic, paper, and paint industries [24–26]. One material that has been suggested as a good replacement for fly ash is low-grade clay (clay with relatively lower kaolinite content). Several researchers [27–31] have reported interesting results after studying the effect of this type of clay in cementitious systems. Given its low cost and extensive availability, it makes an excellent starting material for the synthesis of geopolymers.

Tahmasebi et al. [32] studied the potential use of low-reactivity calcined clays for synthesizing geopolymer. The clays were selected based on properties such as particle fineness, amorphous nature, and oxide composition. The optimum characteristics were achieved at 550 °C calcining temperature. Geopolymer concrete synthesized with the uncalcined clay resulted in a 7-day compressive strength of 31.7 MPa, which increased by 33.7% when it was synthesized with the calcined clay. The geopolymer concrete, however, suffered a reduction when the curing period was extended to 28 days. This reduction in strength was attributed to the creation of pores and microcracks due to high-temperature curing. Hamdi et al. [33] also reported an improvement in compressive strength when clay calcined at 850 °C was used as the source material in the preparation of geopolymer concrete. Three concentrations of NaOH were prepared and used as the alkali activation. Ogundiran and Kumar [34] also investigated the potential use of Nigerian clay as the main precursor in geopolymer and utilized NaOH and $NaOH/Na_2SiO_3$ as the alkali solutions. Calcined clays synthesized with NaOH resulted in relatively lower compressive strength values as compared to $NaOH/Na_2SiO_3$. Investigating the effect of alkali type and concentration on calcined clay mortar, Bature et al. [35] observed a reduction in strength in $Na_2SiO_3$-activated calcined clay mortar cured at ambient temperature. This strength reduction was due to a weakening of the alkali solution caused by chemically bound water. Ferone et al. [36] also studied the potential synthesis of low-reactive calcined clay sediments as a geopolymer binder. They observed a significant improvement in the reactivity of the source material and a consequential increase in compressive strength when the calcining temperature was increased from 400 °C to 750 °C.

In the production and promotion of geopolymer concrete, availability of aluminosiliceous materials is of primary importance. Several studies over the years have focused on the sole use of fly ash in geopolymer systems. However, fly ash is an industrial by-product that can only be found in specific regions. Low-grade clays, on the other hand, are available almost everywhere but have not received sufficient attention to their potential use as a precursor for geopolymer synthesis. The findings of this research would contribute to forming a roadmap for the utilization of low-reactivity clays in geopolymer systems, especially in regions where high-grade kaolinitic clays are unavailable.

In this research, low-grade calcined clay is used to partially substitute fly ash in the preparation of geopolymer mortar with the intention of using it for construction applications. Earlier investigations focused on either using fly ash or calcined clay as the source materials for geopolymer synthesis. The few studies conducted into the potential use of calcined clay for geopolymer synthesis focused on the utilization of pure kaolinitic clays, which possess excellent reactivity. This present study focuses on the use of low reactive clay at varying compositions to partially replace fly ash in the synthesis of a geopolymer binder. The combined effect of the two source materials on compressive strength, setting time, shrinkage, and porosity. Isothermal calorimetry was used in characterizing the effect of calcined low-grade calcined clay on the geopolymerization process. XRD analysis was conducted to determine the phase composition of the 28-day cured geopolymer paste.

## 2. Materials and Methods

### 2.1. Materials

The main precursors used for this study are aluminosiliceous fly ash and low-grade calcined clay. The fly ash, obtained from Lafarge Tarmac, had a specific gravity of 2.3 g/cm$^3$ and a specific surface area of 1605 m$^2$/kg. The calcined clay was produced by calcining clay with a low kaolinite content (19%) at 750 °C. Calcination time was set for 2 h at a heating rate of 10 °C/min. The calcined clay was removed from the kiln and left to cool to ambient temperature on a laboratory bench. The specific gravity and specific surface area of the calcined clay were measured to be 2.2 g/cm$^3$ and 1689 m$^2$/kg respectively. Reagent-grade sodium hydroxide (NaOH) with 98% purity and sodium silicate (Na$_2$SiO$_3$) were mixed and used in the alkali solution preparation. A 12 M NaOH was prepared by dissolving 99% of the pellets in water. The preparation was done 24 h ahead of its usage. The SiO$_2$ and Na$_2$O content of the sodium silicate were 16.84% and 28.51%, respectively. Its SiO$_2$/Na$_2$O molar ratio was 2.0. Sharp sand passing through 4.7 mm and retaining on a 0.6 mm sieve size was used as the fine aggregate.

### 2.2. Methods

#### 2.2.1. Mix Proportion

The sand and geopolymer binder (fly ash + calcined clay + alkali activator) were used to formulate 6 different mixes comprising varying fly ash-to-calcined clay ratios, thus, 100:0 (CC0), 80:20 (CC20), 60:40 (CC40), 40:60 (CC60), 20:80 (CC80), and 0:100 (CC100). Mixes were designed with a sand-to-binder ratio of 1.5, alkali solution-to-binder ratio of 0.5, and water-to-binder ratio of 0.4.

#### 2.2.2. Geopolymer Mortar Preparation

During preparation of the geopolymer mortar, the paste was first prepared by mixing the fly ash and calculated quantity of calcined clay before adding the alkali activator and allowing mixing for 5 min. The sand was added to the whole mixture and allowed to mix for an additional 5 min to ensure consistency and workability. The mortar was placed in 50 × 50 × 50 mm stainless steel molds and compacted on a vibrating table to ensure uniform particle arrangement and to get rid of air bubbles and voids. The specimens were stored in a controlled environment with a temperature of 20 °C and relative humidity of 80%. The specimens were demolded after 48 h and subjected to thermal curing at a temperature of 120 °C. Temperature selection was adopted based on results obtained by previous researchers, which suggest a minimum curing temperature of 120 °C for geopolymer mortars involving impure calcined kaolinitic clays if structural integrity is to be achieved [37,38]. The specimens, before packing in the oven, were wrapped in oven bags to minimize water loss due to thermal curing. The samples were taken out of the oven after 24 h and kept at room temperature pending the testing day. Figure 1 is a flow diagram of the geopolymer preparation.

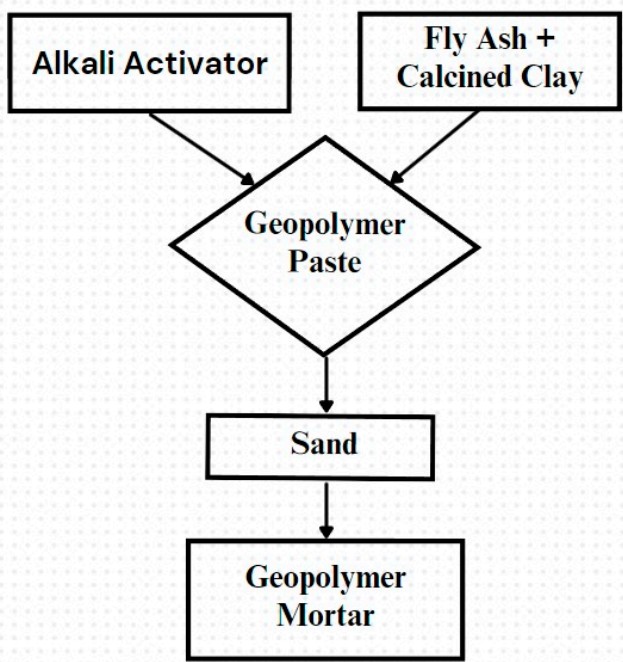

**Figure 1.** Flow diagram of geopolymer mortar preparation.

### 2.2.3. Testing Methods

The elemental composition of the calcined clay and fly ash samples was determined using the X-ray fluorescence (XRF) technique. Microstructural studies of the precursors were done using the scanning electron microscope (SEM). The Vicat method was used to evaluate the setting time of the fly ash-calcined clay geopolymer paste according to methods prescribed by ASTM C191-21 [39]. The compressive strength of the mortar cubes, according to ASTM C109/C109M-20 [40], was measured after 7-, 28-, and 91-day curing. Determination of relevant phases in the geopolymer mortar specimens was carried out by a 3rd generation Malvern Panalytical Empyrean equipped with multicore optics. Readings were taken from 10 to 80 2θ degrees. Particle size distribution of the fly ash and calcined clay was determined by laser diffraction. The pore structure of the 28-day cured specimens at microscopic levels were determined using the Mercury Intrusion Porosimetry technique (MIP).

The Fourier Transformed Infrared technique was used as a complementary tool to XRD to determine the mineralogy of the precursors. A thermometric TAM air conduction calorimeter was used to track the rate of heat evolution. In this test, 4 mL of the alkali solution (stored in the injector) and 5 g of the pre-mixed precursor was kept in the reaction ampoule for 300 min to avoid the possibility of heat turbulence. The alkali solution was injected into the powdered precursor and mixed uniformly for 2 min. The attached computer automatically generated heat flow data continuously for up to 72 h. The shrinkage test was conducted in accordance with ASTM C490 [41]. Mortar bars were prepared using 75 × 75 × 285 mm molds. After demolding, the specimens were covered with plastic sheets to avoid loss of water. The length of the specimens was taken using a length comparator and stored in a controlled environment with a relative humidity of 50% and temperature of 23 ± 4 °C. Subsequent measurements were taken after 1, 3, and 7 days. Measurements were continuously taken at 7-day intervals until 91 days. The specimens were prepared in triplicate, and an average shrinkage was calculated as recommended by ASTM C490 [41].

## 3. Results and Discussion

### 3.1. Raw Materials Characterization

FTIR is a technique used to study the chemical bonds present in the functional groups of cementitious materials. The FTIR spectra of the low-grade calcined clay and fly ash are shown in Figure 2. Bands within the range of 2500 and 2000 cm$^{-1}$ were assigned to OH stretching of inner surface hydroxyl groups, whereas bands appearing between 1000 and 1200 cm$^{-1}$ were assigned to vibrations within the OH group as a result of water absorption [5]. The band 780 cm$^{-1}$ on the calcined clay corresponds to Si-O stretching. The 850 cm$^{-1}$ absorption band present in the fly ash, assigned to the coupled Al-O and Si-O bending, could not be seen in the calcined clay due to the differences in the elemental and mineralogical composition of the two materials [42].

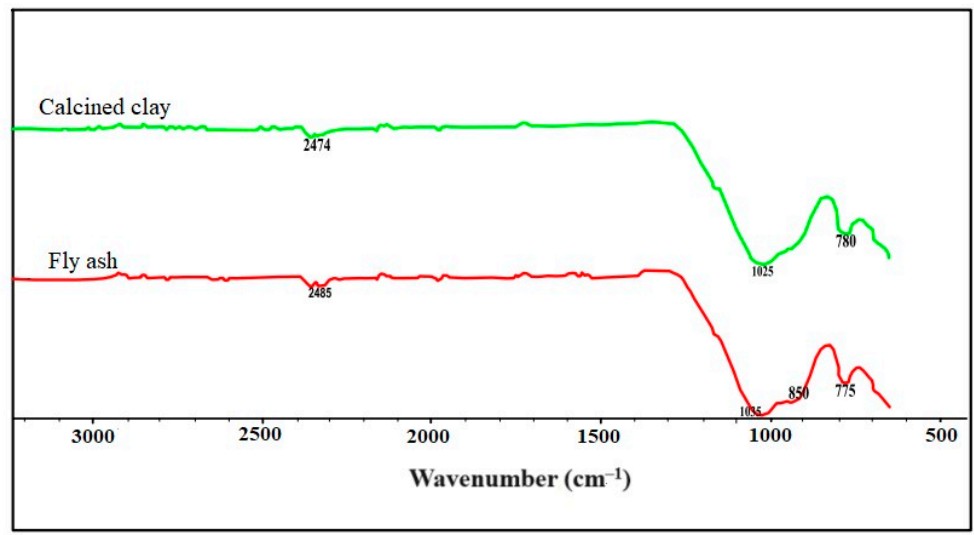

**Figure 2.** FTIR spectra of calcined clay and fly ash.

Table 1 shows the oxide composition of the starting precursors. SiO$_2$ and Al$_2$O$_3$ are the dominant oxides present in both precursors. The SiO$_2$ content in the calcined clay was determined to be 7.4% more than that of fly ash. Al$_2$O$_3$ content was about 27.5% higher in fly ash than calcined clay. However, the two materials possessed enough quantities of oxides for reactivity.

**Table 1.** Chemical composition of calcined clay, fly ash, and OPC.

| Composition, % | SiO$_2$ | Al$_2$O$_3$ | Fe$_2$O$_3$ | MgO | CaO | Na$_2$O | K$_2$O | MnO | TiO$_2$ | P$_2$O$_5$ | SO$_3$ | LOI |
|---|---|---|---|---|---|---|---|---|---|---|---|---|
| Calcined clay | 62.77 | 18.71 | 11.68 | 1.89 | 0.25 | 0.03 | 2.12 | 0.46 | 0.41 | 0.03 | 0.19 | 1.46 |
| Fly ash | 58.1 | 25.82 | 4.25 | 0.25 | 2.28 | - | 1.13 | | 1.1 | 0.22 | 0.12 | 6.73 |

SEM analysis, shown in Figure 3, revealed the spherical-shaped nature of the fly ash particles, whereas the calcined clay showed a thick angular flaky nature due to the destruction of the kaolinitic plate [43]. XRD patterns of the fly ash and calcined clay, presented in Figure 4, showed the presence of quartz (SiO$_2$), mullite (3Al$_2$O$_3$·2SiO$_2$), anatase (TiO$_2$) and calcite (CaCO$_3$), which is indicative of the crystalline minerals. Particle size distribution (PSD) of the precursors is critical to compressive strength development of the resultant geopolymer mortar. The fly ash had a d10, d50, and d90 particle sizes finer than 1.8, 6.0, and 18.2 μm, respectively, and the calcined clay, finer than 3.2, 12.3, and 73.1 μm, respectively. The PSD of the fly ash and calcined clay are presented in Figure 5.

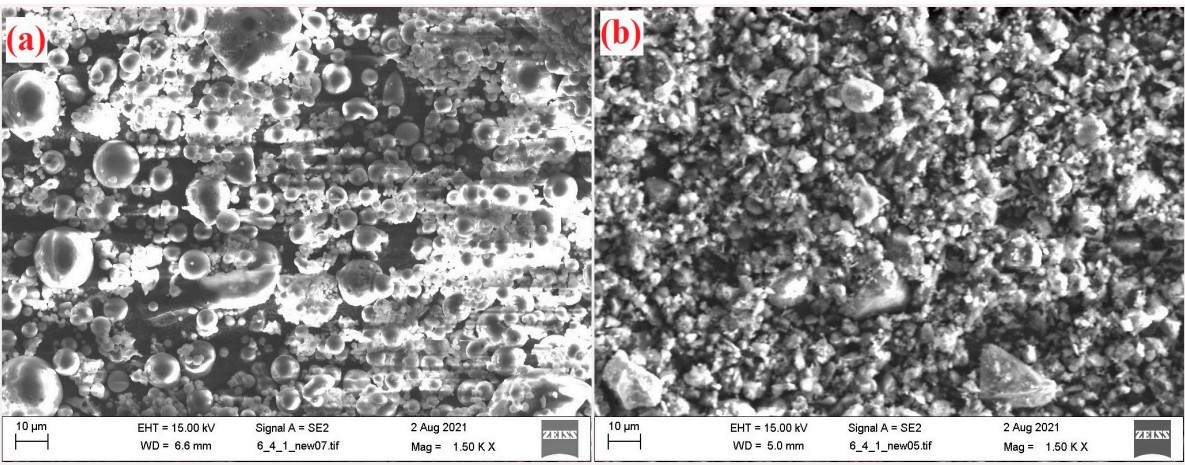

**Figure 3.** SEM image of (**a**) fly ash and (**b**) calcined clay.

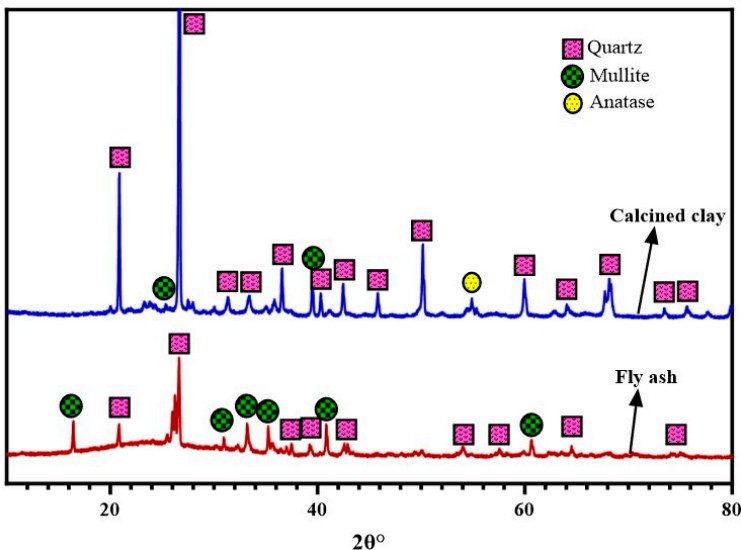

**Figure 4.** XRD spectra of fly ash and calcined clay.

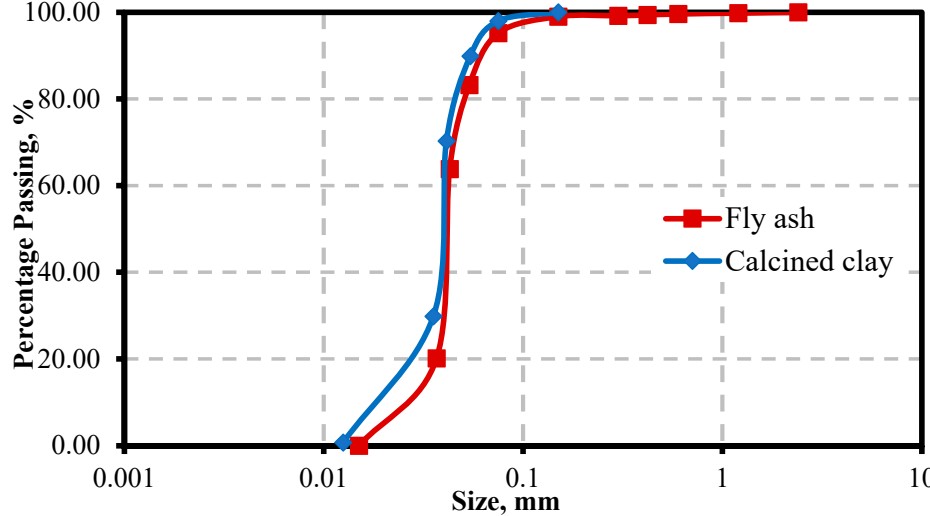

**Figure 5.** Particle size distribution of fly ash and calcined clay.

*3.2. Isothermal Calorimetry*

As shown in Figures 6 and 7, isothermal calorimetry was used in characterizing the influence of low-grade calcined clay on the geopolymerization process. The rate of heat evolution is monitored up to 72 h. It is observed from Figure 6 that heat evolution decreased (28.2 to 20.5 mW/g) with increasing calcined clay content in the geopolymer paste. This trend is consistent with results obtained by Zhang et al. [44] and Ogundiran and Kumar [45], who reported the highest rates of heat evolution between 28.4 and 30 mW/g in geopolymer pastes prepared with metakaolin. It is obvious from Figure 6 that the partial substitution of fly ash with calcined clay reduces the initial rate of heat evolution. The incorporation of calcined clay slows down the formation of the first exothermic peak and makes it less intense [44]. This outcome is unexpected because it was not anticipated that a simple substitution of the fly ash with a low reactive impure calcined clay would result in a considerable slowdown of the fly ash dissolution process. Therefore, it is possible that the polymerization of the $SiO_2$ and $Al_2O_3$ that dissolve from the fly ash also contributes significantly to the first exothermic peak [44]. If so, the partial substitution of fly ash with calcined clay may cause a movement of this peak by increasing the supersaturation period and formation of gel. This is because, under geopolymerization conditions, the dissolution of calcined clay is slower than fly ash [46].

The cumulative heat released by the system within the initial 72 h of geopolymerization reaction is shown in Figure 8. The addition of calcined clay causes a decrease in the cumulative heat, decreasing with increasing calcined clay content. From Figures 6 and 7, the geopolymerization of the system containing only fly ash (CC0) emitted more heat compared to that prepared with only calcined clay (CC100). This is attributed to the variations in reactivity of the two starting materials. The reactivity of the calcined clay is slightly lower than the fly ash due to the impure nature of the clay (with a kaolinite content of less than 20%) used for its preparation.

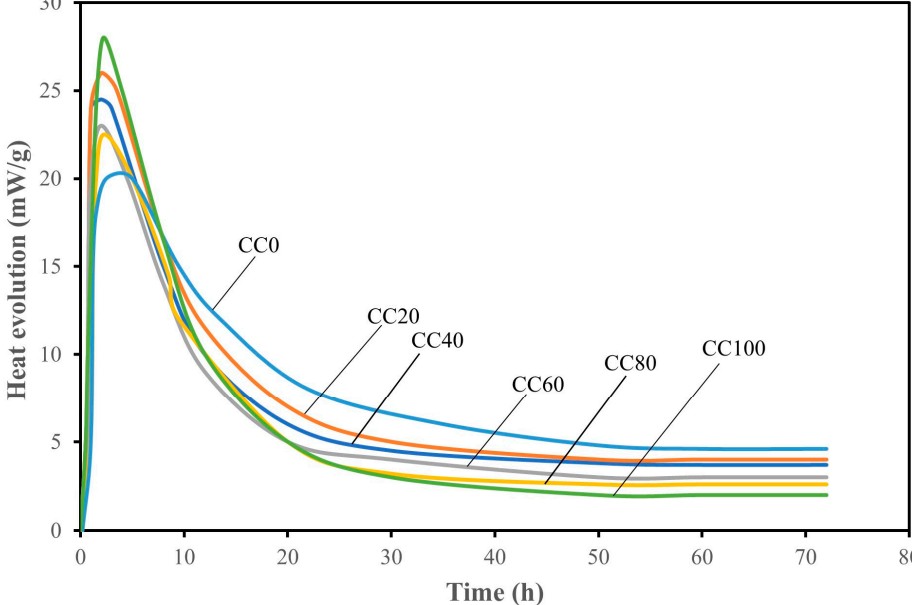

**Figure 6.** Effect of calcined clay on heat evolution.

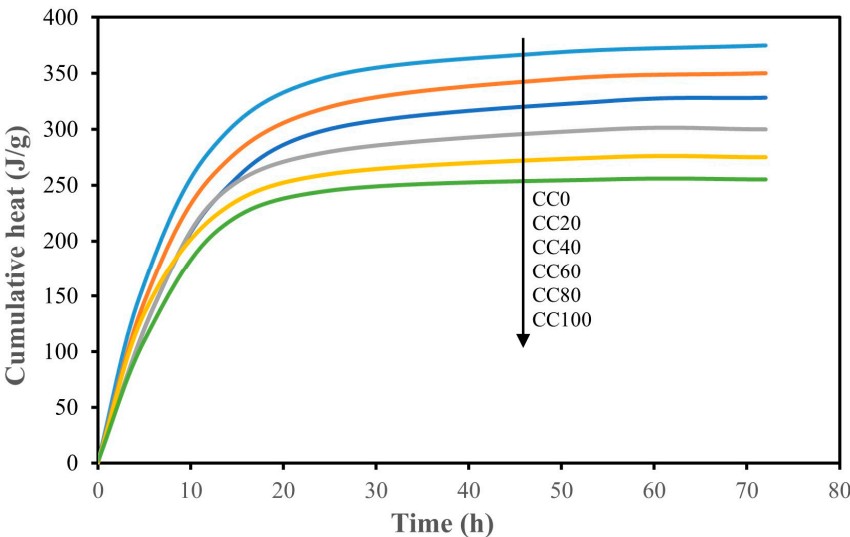

**Figure 7.** Effect of calcined clay on cumulative heat.

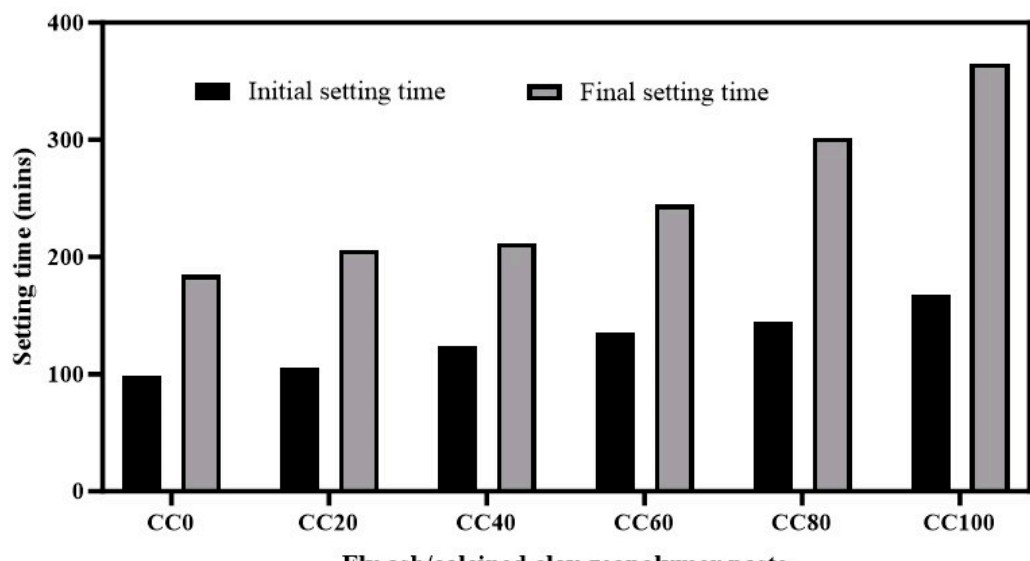

**Figure 8.** Effect of calcined clay on setting time of geopolymer mortar.

### 3.3. Setting Times

Setting time is the process by which cement loses its plasticity and becomes dense at the onset of hydration. The setting times of the fly ash/calcined clay geopolymer pastes are presented in Figure 8. Initial and final setting times recorded for geopolymer paste containing fly ash (CC) were 98 min and 185 min, respectively. These setting times increased monotonically with increasing calcined clay content. The addition of 20%, 40%, 60%, 80% and 100% by mass fraction of calcined clay increased the initial setting time by 6.7%, 21%, 27.4%, 32.4%, and 41.7%, respectively, whereas the final setting time increased by 10.2%, 27%, 24.5%, 38.7%, and 49.3%, respectively. This increase is consistent with results obtained from the isothermal calorimetry. It is clear from the setting time and heat evolution results that the disintegration of the precursors to form a gel prolongs, even after setting is achieved [33]. The increase in setting time with increasing calcined clay content means that, in practice, it will take a relatively longer period for the geopolymer concrete to achieve a minimum level of stiffness.

### 3.4. Compressive Strength

Figure 9 presents the compressive strength data of the geopolymer mortar at 7, 28, and 91 days. At 7 days, compressive strength values were generally low. They ranged between 31 and 41.33 MPa. The specimen marked CC20 recorded the highest strength at 7 days. A similar trend in strength development was observed at 28 days. The compressive strength of the specimen containing 20% calcined clay (CC20) at 91 days increased by 5.6%, 17%, and 18.5%, respectively, while CC40 showed strength reductions of 5.9%, 5.6%, and 2.3%, respectively. As calcined clay content increased in the geopolymer mix, the compressive strength continuously decreased. Comparing the results of specimens prepared with only calcined clay (CC100) and fly ash (CC0), it was observed that the strength of CC0 was superior to CC100 by 12.7%, 6.2%, and 4.9%, respectively. However, all specimens achieved a fairly high strength, and from 7 to 91 days, all specimens significantly increased in strength. It is worth noting that the strength results closely match the calcined clay reactivity during the first 72 h, as observed in the calorimetry. The differences in strength development suggest that the replacement of fly ash with calcined clay affects the nature of the gel formed after dissolution and could lead to pore structure and microstructural changes [35]. The optimum strength was recorded when 20% calcined clay was added to the mix. This is consistent with earlier reports. According to Duan et al. [46], the incorporation of 20% metakaolin into fly ash-based geopolymer concrete significantly improved compressive strength. Parashar et al. [47] also reported improved strength when geopolymer was synthesized with calcined clay.

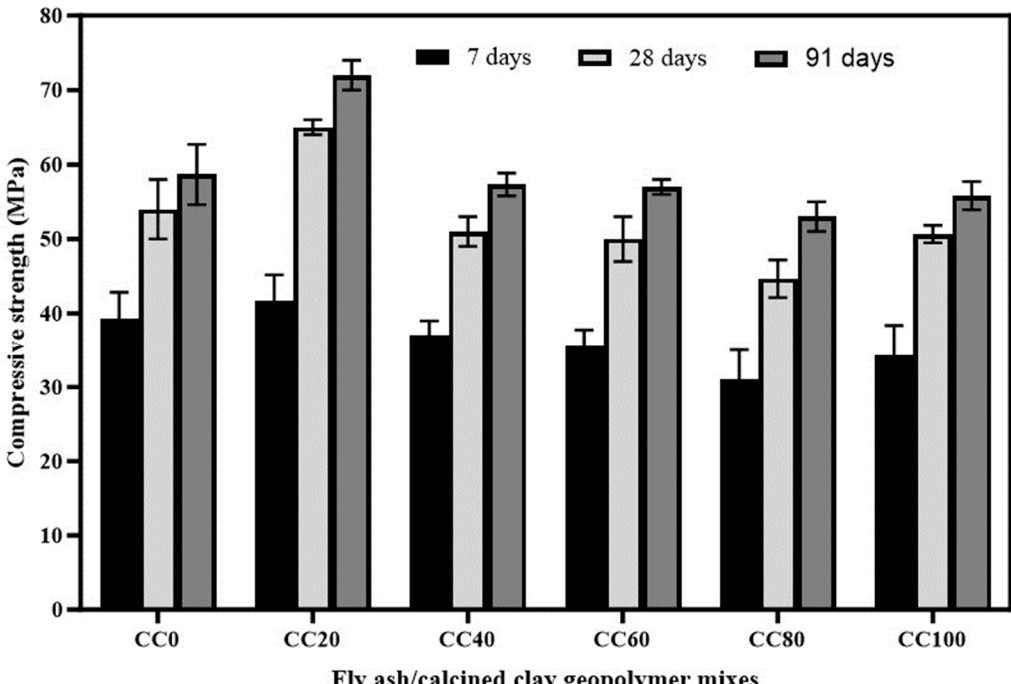

**Figure 9.** Compressive strength of geopolymer mortar containing calcined clay.

### 3.5. Mercury Intrusion Porosimetry (MIP)

Porosity measurement is an effective technique in the determination of the pore structure of cementitious materials at the microscopic level [48]. The determination of porosity is crucial to the strength and durability of concrete structures. Figure 10 shows the porosity and average pore diameter data determined by MIP. After 28 days of curing, porosity was found to decrease as calcined clay content increased in the geopolymer mix. The highest porosity (24%) was recorded for mixes containing only fly ash (CC0), whereas the calcined clay-based geopolymer mortar (CC100) obtained the lowest porosity (6.5%). Porosity values recorded for CC20, CC40, CC60, and CC80 were 18.45%, 16.54%, 12.65%,

and 11.24% respectively. All mixes that contained calcined clay, irrespective of the level of replacement, were found to be less porous than CC0. This is due to the large surface area of the calcined clay, as demonstrated by the particle size distribution results shown in Figure 5. Again, as shown by the SEM in Figure 3, the calcined clay particles appear to be denser and interlocking as compared to that of fly ash. As a result, the calcined clay particles can significantly contribute to filling in the spaces between fly ash particles, which results in a denser pore structure in blended geopolymer paste. Following the porosity results, geopolymer mixes prepared with fly ash and calcined clay showed a decreasing order of average pore diameter. CC100 and CC0 recorded the lowest and highest average diameters, respectively. Pore diameters for the specimens were 0.018, 0.0179, 0.0165, 0.0158, 0.0132, and 0.0134 μm, respectively. Similar results were reported by previous researchers [48,49].

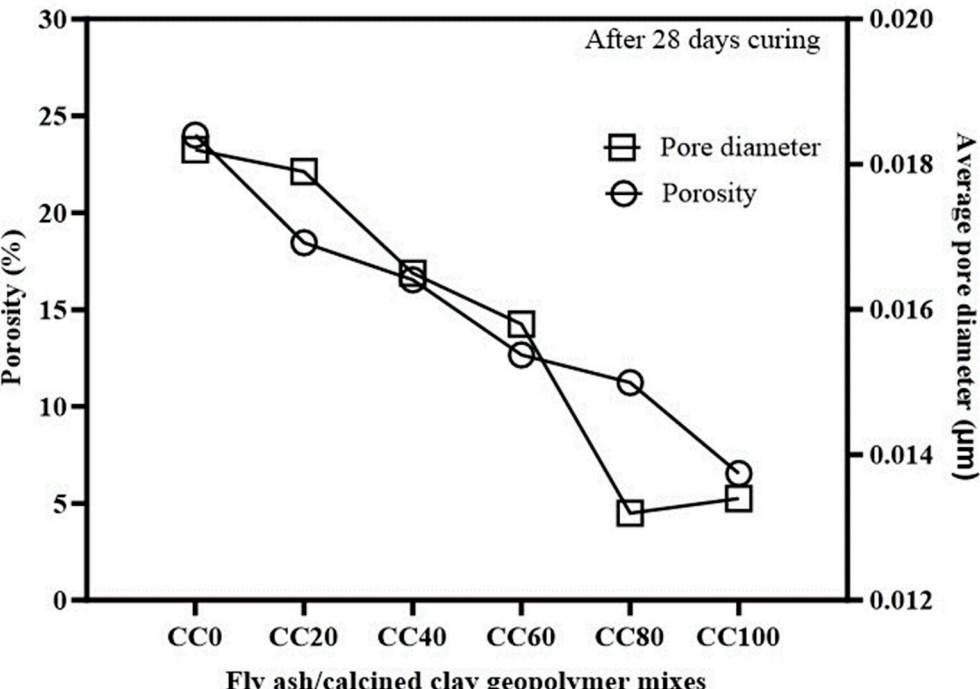

**Figure 10.** Porosity and average particle diameter of fly ash-calcined clay geopolymer.

*3.6. Autogenous Shrinkage*

Autogenous shrinkage was measured to determine the decrease in volume of the geopolymer due to internal drying [50]. During the geopolymerization process, water is removed from the pores through capillarity to form an air-water mixture. This causes pressure to be generated within the capillaries of the binder. The shrinkage properties of fly ash/calcined clay geopolymer mortar are presented in Figure 11. Autogenous shrinkage decreased with increasing calcined clay content, ranging from $523 \times 10^{-6}$ mm/mm to $250 \times 10^{-6}$ mm/mm. Shrinkage was observed to be less severe in mixes containing calcined clay. This means that the samples suffered expansion, which could be due to the rise in internal temperature at early ages because of the high concentration of $Na_2SiO_3$ and NaOH. Shrinkage in calcined clay specimens began after 28 days and consistently increased as the days increased up to 91 days. Among the calcined clay specimens, the least shrinkage value was recorded for the sample marked C100, whereas the highest was recorded in CC0, after 91 days. Generally, shrinkage decreased with increasing calcined clay content. The MIP analysis revealed that refinement of the pore structure of geopolymer specimens containing calcined clay resulted in the release of tensional forces within the pore fluid. The autogenous shrinkage of the geopolymer paste developed in two phases: expansion at the initial stages, which led to a consistent increase in internal strain and finally resulted in shrinkage of the paste. Yang et al. [51], after partially substituting fly ash in metakaolin-based geopolymer, reported a similar trend where the initial stages of

curing experienced expansion but suffered autogenous shrinkage at later ages of curing. This is attributed to the movement of pore fluid into spaces where extreme internal drying occurs due to geopolymerization. This movement of fluid eventually causes an increase in the internal relative humidity, which reduces the capillary stresses, thereby causing shrinkage [50].

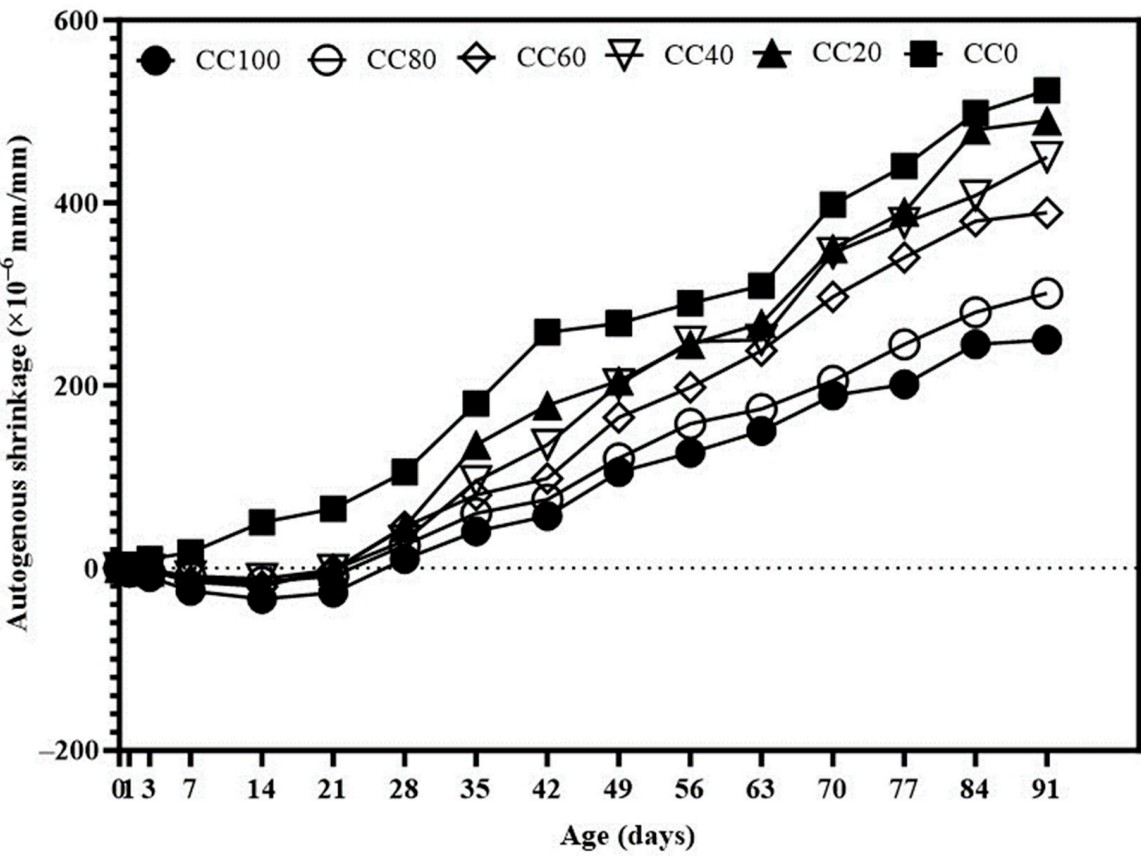

**Figure 11.** Effect of calcined clay on autogenous shrinkage.

*3.7. X-ray Diffraction Analysis*

XRD analysis was conducted to determine the phase composition of the 28-day cured geopolymer paste and is presented in Figure 12. The main phase appearing in all specimens was quartz, which is mostly as a result of geopolymerization reaction [52,53]. Other minerals present include mullite, anorthite, calcite, kaolinite, and tobermorite. Average size peaks observed within 20 and 40 $2\theta°$ suggest the production of a greater amount of amorphous reaction product, which is most likely due to the sodium alumino-silicate hydrate gel [54,55]. Additionally, medium to microscopic peaks between 35 and 50 $2\theta°$ were visible, which might indicate the development of weakly crystalline CASH gel phases [56,57]. The quartz and mullite phases are represented in the diffraction patterns by a significant number of sharp peaks. Similar peak positions and intensities were observed in all blended fly ash-calcined clay geopolymer pastes. Another notable phase present was calcite, found between 35 and 40 $2\theta°$. Mullite, as expected, was also prominent among the phases present. The mullite peaks appear to have reduced or vanished as the calcined clay content increased. Anorthite phases were also observed between 20 and 30 $2\theta°$. Tobermorite phase was seen at 29 $2\theta°$. The presence of tobermorite is essential for its contribution to the hardening of the geopolymer paste [58].

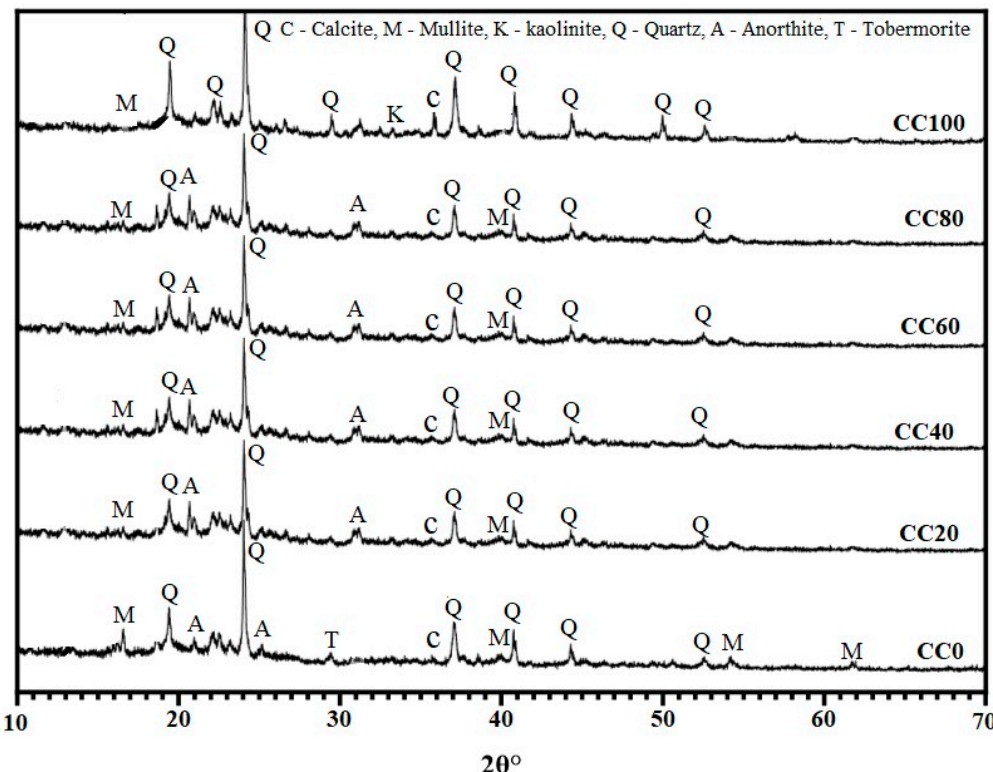

**Figure 12.** XRD diffraction of fly ash-calcined clay geopolymer cured for 28 days.

### 4. Conclusions

This research has investigated the influence of low-grade calcined clay on the fresh, hardened, and mineralogical characteristics of fly ash-based geopolymer mortar synthesized with a single liquid-to-solid ratio. From this study, the mutual reliance on the physical and inherent properties of the two precursors to produce geopolymer mortar with desirable properties has been shown. Variations in the reactivities of the fly ash and calcined clay contributed greatly to the findings of this study, which are listed below:

- As revealed by the isothermal conduction calorimetry results, heat evolution decreased with increasing calcined clay content. The addition of calcined clay reduced the surface interaction between the dissolved particles in the alkali solution, leading to slow initial reactivity.
- The incorporation of 20% calcined clay in the mix favored early and late compressive strength development, outperforming the reference geopolymer mortar (containing 100% fly ash) by 5.6%, 17%, and 18.5% at 7, 28, and 91 days, respectively. Geopolymer mixes containing 100% fly ash, however, obtained slightly improved compressive strength figures at all curing ages as compared to the ones prepared with 100% calcined clay. Setting times increased with increasing calcined clay content in the geopolymer mix. Optimum replacement was found to be 20%.
- The highest porosity (24%) was recorded for mixes containing only fly ash (CC0), whereas the calcined clay-based geopolymer mortar (CC100) obtained the least porosity (6.5%). All mixes that contained calcined clay, irrespective of the level of replacement, were found to be less porous than CC0. The calcined clay particles appeared to be denser and interlocking as compared to that of fly ash.
- Shrinkage was observed to be less severe in mixes containing calcined clay. The MIP analysis revealed that refinement of the pore structure of geopolymer specimens containing calcined clay resulted in the release of tensional forces within the pore fluid.

This work has studied the potential use of calcined clay to partially substitute fly ash for geopolymer synthesis. Its effect on setting time, shrinkage, compressive strength devel-

opment, and porosity has been revealed. Further studies, however, should be conducted into the impact of microstructural changes on these properties. Other areas of this research that also require attention are durability measurements and life cycle assessment in order to optimize the use of calcined clay in geopolymer systems.

**Author Contributions:** Conceptualization, M.K.; methodology, M.K. and K.B.; software, M.K. and K.B.; validation, M.K.; formal analysis, M.K. and K.B.; investigation, K.B. and M.K.; resources, M.K. and K.B.; data curation, K.B. and M.K.; writing—original draft preparation, K.B.; writing—review and editing, M.K.; visualization, M.K.; supervision, M.K.; project administration, M.K.; funding acquisition, M.K. and K.B. All authors have read and agreed to the published version of the manuscript.

**Funding:** This research was funded by Coventry University under project code 17172-01 and the APC was funded by MDPI open access publishing in Basel/Switzerland.

**Institutional Review Board Statement:** Not applicable.

**Informed Consent Statement:** Not applicable.

**Data Availability Statement:** Not applicable.

**Conflicts of Interest:** The authors declare no conflict of interest.

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
