# Peer review of "Impact of Low-Reactivity Calcined Clay on the Performance of Fly Ash-Based Geopolymer Mortar"

_sustainability, doi:10.3390/su151813556_

Round 1
Reviewer 1 Report
This paper is well-written and the results are good. More discussions are needed in the result section to improve the quality of the paper
well written
Author Response
COMMENTS
- This paper is well-written and the results are good. More discussions are needed in the result section to improve the quality of the paper.
More information has been added in the discussion of the results, as suggested. For instance, additions made to section 3.6 has been shaded green below.
“Autogenous shrinkage was measured to determine the decrease in volume of the geopolymer due to internal drying [47]. During the geopolymerization process, water is removed from the pores through capillarity to form an air-water mixture. This causes pressure to be generated within the capillaries of the binder. The shrinkage properties of fly ash/calcined clay geopolymer mortar is presented in Figure 11. Autogenous shrinkage decreased with increasing calcined clay content, ranging from 523×10-6 mm/mm - 250×10-6 mm/mm. Shrinkage was observed to be less severe in mixes containing calcined clay. This means that the samples suffered expansion which could be due to rise in internal temperature at early ages because of the high concentration of Na2SiO3 and NaOH. Shrinkage in calcined clay specimens began after 28 days and consistently increased as the days increased up to 91 days. Among the calcined clay specimens, the least shrinkage value was recorded for the sample marked C100 whereas the highest was recorded in CC0, after 91 days. Generally, shrinkage decreased with increasing calcined clay content. The MIP analysis revealed that refinement of the pore structure of geopolymer specimens containing calcined clay resulted in the release of tensional forces within the pore fluid. The autogenous shrinkage of the geopolymer paste developed in two phases; expansion at the initial stages, which led to a consistent increase in internal strain and finally resulted in shrinkage of the paste. Yang et al. [48], after partially substituting fly ash in metakaolin-based geopolymer reported a similar trend where the initial stages of curing experienced expansion but suffered autogenous shrinkage at later ages of curing. This is attributed to the movement of pore fluid into spaces where extreme internal drying occurs due to geopolymerization. This movement of fluid eventually causes an increase in the internal relative humidity which reduces the capillary stresses and thereby causing shrinkage [49].”
Reviewer 2 Report
- How does the incorporation of calcined clay affect the geopolymerization process and resultant mortar?
- What are the specific ratios of fly ash to calcined clay used in the six different mortar mixes?
- How does the addition of calcined clay impact the initial reactivity and heat evolution of the geopolymer paste?
- What are the implications of the changes in setting times observed with increasing calcined clay content?
- Explain the trend observed in compressive strength development with varying levels of calcined clay replacement.
- How does the porosity of the geopolymer mortar change with different levels of calcined clay substitution?
- What microstructural changes contribute to the observed differences in autogenous shrinkage between fly ash and calcined clay mixes?
- Describe the mineralogical phases identified through X-ray diffraction analysis in the geopolymer mortar.
- Why is the 20% calcined clay replacement considered optimal for enhancing the compressive strength of the geopolymer mortar?
- Compare the heat evolution patterns in geopolymer pastes containing fly ash with those containing calcined clay.
- How do the findings from this study contribute to the understanding of the interaction between fly ash and calcined clay in geopolymerization?
- What practical implications do the changes in setting times have for the production of geopolymer-based materials?
- Discuss the potential benefits and challenges of using low-reactivity calcined clay in geopolymer formulations.
- How can the observed changes in pore structure and porosity be explained based on the characteristics of fly ash and calcined clay?
- What recommendations can be drawn from this research for optimizing the performance of fly ash-based geopolymer mortars in practical applications?
Minor editing of English language required
Author Response
The authors wish to express their deep-seated appreciation to the reviewer for the constructive and insightful feedback. All corrections and suggestions are duly implemented, as directed. Responses to all comments have been outlined below:
- How does the incorporation of calcined clay affect the geopolymerization process and resultant mortar?
A summary of the effect of calcined clay on the geopolymerization process is seen in the conclusion, thus, reduction of surface interaction between the dissolved particles in the alkali solution leading to slow initial reactivity, increase in setting time, decreased porosity, decreased autogenous shrinkage, improved compressive strength at early and ultimate age.
- What are the specific ratios of fly ash to calcined clay used in the six different mortar mixes?
The fly ash-to-calcined clay ratio is seen under Section 2.2.1 (Line 131-35).
“The sand and geopolymer binder (fly ash + calcined clay + alkali activator) were used to formulate 6 different mixes comprising of varying fly ash-to-calcined clay ratios, thus, 100:0 (CC0), 80:20 (CC20), 60:40 (CC40), 40:60 (CC60), 20:80 (CC80), and 0:100 (CC100)”.
- How does the addition of calcined clay impact the initial reactivity and heat evolution of the geopolymer paste?
Due to the low kaolinite content of the clay, the addition of calcined clay reduced the surface interaction between the dissolved particles in the alkali solution, leading to slow initial reactivity. The dissolution of the calcined clay under geopolymerization conditions is quite slow, and consequently delays the heat evolved from the geopolymer reaction. This is mentioned under Section 3.2 (Line 232-240).
- What are the implications of the changes in setting times observed with increasing calcined clay content?
Since time is of great essence in construction, setting time is crucial for the planning of geopolymer concrete projects. The increase in setting time with increasing calcined clay content means that, in practise, it will take a relatively longer period for the geopolymer concrete to achieve a minimum level of stiffness or strength. This has been inserted in Section 3.3 (Line 265-267). It reads:
“The increase in setting time with increasing calcined clay content means that, in practice, it will take a relatively longer period for the geopolymer concrete to achieve a minimum level of stiffness.”
- Explain the trend observed in compressive strength development with varying levels of calcined clay replacement.
From Figure 9, increase in calcined clay replacement beyond 20% resulted in a decrease in compressive strength at all curing ages, as compared to fly ash. This is due to the low reactivity of the calcined clay. The low reactivity is connected to the low level of kaolinite in the clay (19%, indicated under section 2.2, line 119). This could also be due to “dilution effect” caused by over substitution with calcined clay.
- How does the porosity of the geopolymer mortar change with different levels of calcined clay substitution?
Porosity was found to increase as calcined clay content increased in the geopolymer mix. This is due to the large surface area of the calcined clay as demonstrated by the particle size distribution results shown in Figure 5. Again, as shown by the SEM in Figure 3, the calcined clay particles appear to be denser and interlocking as compared to that of fly ash. This is indicated in line 302-308.
- What microstructural changes contribute to the observed differences in autogenous shrinkage between fly ash and calcined clay mixes?
Unfortunately, this study did not include microstructural analysis of the hydrated geopolymer mortar and so the effect of microstructural changes on autogenous shrinkage could not be ascertained. The authors are grateful to the reviewer for pointing this out. This has been noted and would be considered in future works. This has been highlighted in the recommendation paragraph (after the conclusion).
- Describe the mineralogical phases identified through X-ray diffraction analysis in the geopolymer mortar.
The main phase appearing in all specimens was quartz, which is mostly as a result of geopolymerization reaction. Other minerals present include mullite, anorthite, calcite, kaolinite and Tobermorite. The quartz phase is an indication of geopolymerization reaction, geopolymeric product formation, and the presence of sand in the mortar samples. Another notable phase present was calcite, found between 35 and 40 2θ°. Mullite, as expected, was also prominent among the phases present. The mullite peaks appear to have reduced or vanished as the calcined clay content increased. Anorthite phases were also observed between 20 and 30 2θ°. The presence of Anorthite mineral implies the formation of sodium aluminosilicate hydrate (NASH) and calcium aluminosilicate hydrate (CASH) gel. This has been explained in section 3.7.
- Why is the 20% calcined clay replacement considered optimal for enhancing the compressive strength of the geopolymer mortar?
It is observed from the results that, beyond 20%, compressive strength declined. This means that, over elaboration of the fly ash-calcined clay mixture with calcined clay slows down dissolution of the precursors in solution, thereby leading to reduced strength. Therefore, per this combination (fly ash-calcined clay), the use of lower amounts of calcined clay (up to 20%) generates the best mechanical properties.
- Compare the heat evolution patterns in geopolymer pastes containing fly ash with those containing calcined clay.
The geopolymerization of the system containing only fly ash (CC0) emitted more heat compared to that prepared with only calcined clay (CC100). This is attributed to the variations in reactivity of the two starting materials. This is shown in line 244-247.
- How do the findings from this study contribute to the understanding of the interaction between fly ash and calcined clay in geopolymerization?
This study has shown the superior characteristics of fly ash against low-grade calcined clay in geopolymerization systems. The superior characteristics is due to its physical, chemical and mineralogical properties, resulting in better reactivity. However, the addition of calcined clay, due to its nature, slows down the dissolution process, affecting mechanical and other properties. This has been highlighted in the conclusion.
- What practical implications do the changes in setting times have for the production of geopolymer-based materials?
Since time is of great essence in construction, setting time is crucial for the planning of geopolymer concrete projects. The increase in setting time with increasing calcined clay content means that, in practise, it will take a relatively longer period for the geopolymer concrete to achieve a minimum level of stiffness or strength. This has been inserted in Section 3.3 (Line 265-267).
- Discuss the potential benefits and challenges of using low-reactivity calcined clay in geopolymer formulations.
Since fly ash (a major precursor in geopolymer concrete) has the potential of becoming scarce in the nearest future due to the close-down of coal-fired power plants, it has become necessary to develop hybrid geopolymer mixtures containing fly ash and other precursors. Low-grade clay is particularly of great interest, due to its availability, environmental friendliness and cost effectiveness. However, due to the low reactivity of the calcined clay, which influences other essential properties of the resultant concrete, not too much of it could be utilized in geopolymer formulations, especially when used with fly ash. An excellent impact on strength, shrinkage and porosity could be achieved when replaced up to 20%. The main challenge is that the use of the calcined clay alone (100%) or higher percentages in geopolymer systems may not produce the desired results.
- How can the observed changes in pore structure and porosity be explained based on the characteristics of fly ash and calcined clay?
The particle fineness of the calcined clay was found to be greater than the fly ash, as in the particle size distribution analysis in Fig. 5. This makes the specimens containing more calcined clay more dense and less porous. This is explained in lines 297-311.
“After 28 days of curing, porosity was found to increase as calcined clay content increased in the geopolymer mix. The highest porosity (24%) was recorded for mixes containing only fly ash (CC0) whereas the calcined clay-based geopolymer mortar (CC100) obtained the least porosity (6.5%). All mixes which contained calcined clay, irrespective of the level of replacement, were found to be less porous than CC0. This is due to the large surface area of the calcined clay as demonstrated by the particle size distribution results shown in Figure 5. Again, as shown by the SEM in Figure 3, the calcined clay particles appear to be denser and interlocking as compared to that of fly ash. As a result, the calcined clay particles can significantly contribute to filling in the spaces between fly ash particles, which results in a denser pore structure in blended geopolymer paste. Following the porosity results, geopolymer mixes prepared with fly ash and calcined clay showed a decreasing order of average pore diameter. CC100 and CC0 recorded the least and highest average diameters”.
- What recommendations can be drawn from this research for optimizing the performance of fly ash-based geopolymer mortars in practical applications?
It is recommended that, up to 20% calcined clay should be used in order to achieve the optimum results in fly ash-calcined clay geopolymer systems. Delayed setting should be taken into account when planning for geopolymer construction projects involving fly ash and calcined clays. Further studies into the commercialization, environmental impact and life cycle assessment of fly ash-calcined clay geopolymer concrete should be considered in the future. A paragraph on recommendation has been added (after the conclusion) to reflect this suggestion. It reads:
“This work has studied the potential use of calcined clay to partially substitute fly ash for geopolymer synthesis. Further studies should be conducted into the impact of microstructural changes on shrinkage, porosity and strength development. Other areas of this research that also require attention are durability measurements and life cycle assessment in order to optimise the use of calcined clay in geopolymer systems.”
Reviewer 3 Report
The work is interesting, there is a certain relevance, but it needs to be improved according to the following comments:
1.The summary does not sufficiently reflect the structure of the study. It is desirable to clearly define the relevance, scientific novelty of the study, specify the object and subject of the study.
The essence of the results obtained is presented, but their connection with the problem being solved is not reflected.
It is also desirable to clearly present, due to their features and characteristic differences, these results allowed us to solve this problem. There is no explanation and no practical value of the presented results.
2.Introduction. The current state of knowledge relating to the topic has not been covered and clearly presented, and the authors’ contributions and findings are not emphasized. In this regard, the authors should make their effort to address these issues.
3.The further work should be mentioned at the end of the article. Please modify it.
4. The discussion in the subsections is very sparse. The discussion needs to be improved.
Author Response
The authors wish to express their gratitude to the reviewer for the constructive comments and insightful feedback which, we believe, would undoubtedly enhance the quality of this paper. All corrections and suggestions have therefore been duly implemented, as directed. Responses to all comments have been outlined below:
- The summary does not sufficiently reflect the structure of the study. It is desirable to clearly define the relevance, scientific novelty of the study, specify the object and subject of the study. The essence of the results obtained is presented, but their connection with the problem being solved is not reflected. It is also desirable to clearly present, due to their features and characteristic differences, these results allowed us to solve this problem. There is no explanation and no practical value of the presented results.
The authors are thankful for the suggestion. The abstract has been reviewed and now reads as follows:
“Availability of aluminosiliceous materials is essential for the production and promotion of geopolymer concrete. Unlike fly ash, which can only be found in industrial regions, clays are available almost everywhere, but has not received sufficient attention in its potential use as precursor for geopolymer synthesis. This study investigates the effectiveness of calcined clay as a sole and binary precursor (with fly ash) for the preparation of geopolymer mortar. Fly ash-based geopolymer containing between 0 and 100% low-grade calcined clay were prepared to investigate the effect of calcined clay replacement on the geopolymerization process and resultant mortar, using a constant liquid/solid ratio. Reagent grade sodium hydroxide (NaOH) and sodium silicate (Na2SiO3) were mixed and used for the alkali solution preparation. Six different mortar mixes were formulated using sand and the geopolymer binder, comprising of varying fly ash-to-calcined clay ratios. The combined effect of the two source materials on compressive strength, setting time, autogenous shrinkage and porosity were studied. The source materials were characterized using XRD, SEM, FTIR and XRF techniques. Isothermal calorimetry was used in characterizing the effect of low-grade calcined clay on the geopolymerization process. The addition of calcined clay reduced the surface interaction between the dissolved particles in the alkali solution, leading to slow initial reactivity. Geopolymer mortar containing 20% calcined clay outperformed the reference geopolymer mortar by 5.6%, 17% and 18.5% at 7, 28 and 91 days respectively. The MIP analysis revealed that refinement of the pore structure of geopolymer specimens containing calcined clay resulted in the release of tensional forces within the pore fluid. Optimum replacement was found to be 20%. From this study, the mutual reliance on the physical and inherent properties of the two precursors to produce geopolymer mortar with desirable properties has been shown. The findings strongly suggest that clay containing low content of kaolinite can be calcined and added to fly ash, together with appropriate alkali activators to produce a suitable geopolymer binder for construction applications.
- The current state of knowledge relating to the topic has not been covered and clearly presented, and the authors’ contributions and findings are not emphasized. In this regard, the authors should make an effort to address these issues.
The authors are grateful for this recommendation. The paragraph below presents the current literature in the topic.
“Tahmasebi et al. [32] studied the potential use of low reactivity calcined clays for synthesizing geopolymer. The clays were selected based on properties such as particle fineness, amorphous nature and oxide composition. The optimum characteristics were achieved at 550 °C calcining temperature. Geopolymer concrete synthesized with the uncalcined clay resulted in a 7 days compressive strength of 31.7 MPa, which increased by 33.7% when it was synthesized with the calcined clay. The geopolymer concrete, however, suffered a reduction when curing period was extended to 28 days. This reduction in strength was attributed to the creation of pores and microcracks due to high temperature curing. Hamdi et al. [33] also reported an improvement in compressive strength when clay calcined at 850 °C was used as the source material in the preparation of geopolymer concrete. Three concentrations of NaOH were prepared and used as the alkali activation. Ogundiran and Kumar [34] also investigated the potential use of Nigerian clay as the main precursor in geopolymer and utilized NaOH and NaOH/Na2SiO3 as the alkali solutions. Calcined clays synthesized with NaOH resulted in relatively lower compressive strength values as compared to NaOH/Na2SiO3. Investigating the effect of alkali type and concentration on calcined clay mortar, Bature et al. [35] observed reduction in strength in Na2SiO3-activated calcined clay mortar cured at ambient temperature. This strength reduction was due to a weakening of the alkali solution caused by chemically bound water. Ferone et al. [36] also studied the potential synthesis of low reactive calcined clay sediments as a geopolymer binder. They observed a significant improvement in reactivity of the source material and a consequential increase in compressive strength when the calcining temperature was increased from 400 °C to 750 °C.”
- The further work should be mentioned at the end of the article. Please modify it.
A paragraph on recommendation has been added (after the conclusion) to reflect this suggestion. It reads:
“This work has studied the potential use of calcined clay to partially substitute fly ash for geopolymer synthesis. Its effect on setting time, compressive strength, shrinkage and porosity has been revealed. Further studies, however, should be conducted into the impact of microstructural changes on these properties. Other areas of this research that also require attention are durability measurements and life cycle assessment in order to optimise the use of calcined clay in geopolymer systems.”
- The discussion in the subsections is very sparse. The discussion needs to be improved.
The authors thank the reviewer for this comment. Subsections in the discussion have been improved, as recommended.
Reviewer 4 Report
1. The title "Impact of low-reactivity calcined clay on the performance of fly ash-based geopolymer mortar" is appropriate and effectively summarizes the core focus of the research
2. It would be beneficial if the abstract mentioned how the findings compared with existing literature or what novel insights were brought forward with this study
3. Consider adding a sentence on potential applications or implications of the study's findings for the construction industry or related sectors. This would convey the broader relevance of the research.
4. In the introduction section Consider editing for repetitiveness. For instance, the phrase "low-grade calcined clay" is mentioned multiple times in quick succession.
5. The paragraph starting with “The few reports on” on page 2 lines 85 to 86 needs a revision for clarity purposes
6. The flow diagram presented on page 3 is very poor. Revision requested
7. Please check the Chemical composition of calcined clay, fly ash and OPC. Presented in Table 1 page 5. The sum is not equal 100
8. More intervention is requested for SEM images of fly ash and calcined clay presented in figure 3. Also, more clear images are required. Are they belong to 2021?
9. More results discussion is required in paragraph of Autogenous Shrinkage on page 10.
10. There are a few repetitive statements in the conclusions section, such as mentioning the 20% calcined clay's optimal performance multiple times. Consider streamlining for brevity.
11. Although the conclusion effectively summarizes the study's findings, it could benefit from a brief section discussing potential issues for future research or applications based on these results
The English language used in the research is generally clear and technically sound
Author Response
The authors wish to express their gratitude to the reviewer for the constructive comments and insightful feedback which, we believe, would undoubtedly enhance the quality of this paper. All corrections and suggestions have therefore been duly implemented, as directed. Responses to all comments have been outlined below and the corrected version of the paper based on all four reviwers comments has been attached.
- The title "Impact of low-reactivity calcined clay on the performance of fly ash-based geopolymer mortar" is appropriate and effectively summarizes the core focus of the research.
- It would be beneficial if the abstract mentioned how the findings compared with existingliterature or what novel insights were brought forward with this study.
The authors are thankful for this recommendation. The following few lines have been added to the abstract to reflect this suggestion (Line 26-30).
“From this study, the mutual reliance on the physical and inherent properties of the two precursors to produce geopolymer mortar with desirable properties have been shown. The findings strongly suggest that clay containing low content of kaolinite can be calcined and added to fly ash, together with appropriate alkali activators to produce a suitable geopolymer binder for construction applications.”
- Consider adding a sentence on potential applications or implications of the study's findings for the construction industry or related sectors. This would convey the broader relevance of the research.
The authors are grateful to the reviewer for this insightful suggestion. The following sentences have been added to the introduction (Line 91-98).
“In the production and promotion of geopolymer concrete, availability of aluminosiliceous materials is of primary importance. Several studies, over the years, have focused on the sole use of fly ash in geopolymer systems. However, fly ash is an industrial by-product which can only be found in specific regions. Low-grade clay, on the other hand, can be found almost everywhere but has not received sufficient attention in its potential use as precursor for geopolymer synthesis. The findings of this research would contribute to forming a roadmap for the use of low-reactivity clays in geopolymer systems, especially in regions where high-grade kaolinitic clays are unavailable.”
- In the introduction section Consider editing for repetitiveness. For instance, the phrase "low-grade calcined clay" is mentioned multiple times in quick succession.
The authors thank the reviewer for spotting this repetitiveness. The introduction has been duly edited to correct this error as seen below (Line 64-68).
“One material that has been suggested as a good replacement for fly ash is low-grade clay (clay with relatively lower kaolinite content). Several researchers [27-31] have reported interesting results after studying the effect of low-grade clay this type of clay in cementitious systems. Given its low cost and extensive availability, low-grade clay it makes an excellent starting material for the synthesis of geopolymers.”
- The paragraph starting with “The few reports on” on page 2 lines 85 to 86 needs a revision for clarity purposes.
The sentence has been revised as directed (Line 103-105). It now reads:
“The few studies conducted into the use of calcined clays for geopolymer synthesis have focused on the utilization of pure kaolinitic clays, which possess excellent reactivity.”
- The flow diagram presented on page 3 is very poor. Revision requested.
As directed, the flow diagram has been revised as seen below.
- Please check the Chemical composition of calcined clay, fly ash and OPC. Presented in Table 1 page 5. The sum is not equal 100.
The authors are thankful for this. Admittedly, the oxides in each material in Table 1 do not sum up to 100. The remaining 1.46% in the calcined clay 6.73% in fly ash are the values for loss on ignition. This was deliberately omitted since the emphasis is on the oxides present. This has been inserted as presented below.
|
Composition, % |
SiO2 |
A12O3 |
Fe2O3 |
MgO |
CaO |
Na2O |
K2O |
MnO |
TiO2 |
P2O5 |
SO3 |
LOI |
|
Calcined clay |
62.77 |
18.71 |
11.68 |
1.89 |
0.25 |
0.03 |
2.12 |
0.46 |
0.41 |
0.03 |
0.19 |
1.46 |
|
Fly ash |
58.1 |
25.82 |
4.25 |
0.25 |
2.28 |
− |
1.13 |
− |
1.1 |
0.22 |
0.12 |
6.73 |
- More intervention is requested for SEM images of fly ash and calcined clay presented in figure 3. Also, more clear images are required. Are they belong to 2021?
The authors thank the reviewer for this recommendation. The SEM scans were taken in 2021. This research has been ongoing since 2021 and so some data (especially on the raw materials) were collected in the early stages of the research. Unfortunately, this SEM image is the ones available presently. Clearer images will be obtained for future works.
- 9. More results discussion is required in paragraph of Autogenous Shrinkageon page
The authors are grateful to the reviewer for this suggestion. A few more discussion points have been raised and inserted (Line 317-340) as follows (shown in green):
Autogenous shrinkage was measured to determine the decrease in volume of the geopolymer due to internal drying [47]. During the geopolymerization process, water is removed from the pores through capillarity to form an air-water mixture. This causes pressure to be generated within the capillaries of the binder. The shrinkage properties of fly ash/calcined clay geopolymer mortar is presented in Figure 11. Autogenous shrinkage decreased with increasing calcined clay content, ranging from 523×10-6 mm/mm - 250×10-6 mm/mm. Shrinkage was observed to be less severe in mixes containing calcined clay. It was observed that specimen containing calcined clay recorded negative shrinkage values at early ages up to 21 days. This means that the samples suffered expansion which could be due to rise in internal temperature at early ages because of the high concentration of Na2SiO3 and NaOH. Shrinkage in calcined clay specimens began after 28 days and consistently increased as the days increased up to 91 days. Among the calcined clay specimens, the least shrinkage value was recorded for the sample marked C100 whereas the highest was recorded in CC0, after 91 days. Generally, shrinkage decreased with increasing calcined clay content. After The MIP analysis revealed that refinement of the pore structure of geopolymer specimens containing calcined clay resulted in the release of tensional forces within the pore fluid. The autogenous shrinkage of the geopolymer paste developed in two phases; expansion at the initial stages, which led to a consistent increase in internal strain and finally resulted in shrinkage of the paste. Yang et al. [48], after partially substituting fly ash in metakaolin-based geopolymer reported a similar trend where the initial stages of curing experienced expansion but suffered autogenous shrinkage at later ages of curing. This is attributed to the movement of pore fluid into spaces where extreme internal drying occurs due to geopolymerization. This movement of fluid eventually causes an increase in the internal relative humidity which reduces the capillary stresses and thereby causing shrinkage [49].
- There are a few repetitive statements in the conclusions section, such as mentioning the 20% calcined clay's optimal performance multiple times. Consider streamlining for brevity.
As advised, the second conclusion point has been revised to avoid excessive repetition. It now reads:
“The incorporation of 20% calcined clay in the mix favoured early and late compressive strength development, outperforming the reference mortar (containing 100% fly ash) by 5.6%, 17% and 18.5% at 7, 28 and 91 days respectively. Geopolymer mixes containing 100% fly ash, however, obtained slightly improved compressive strength figures at all curing ages as compared to the ones prepared with 100% calcined clay. Setting times increased with increasing calcined clay content in the geopolymer mix. Optimum replacement was found to be 20%.”
- Although the conclusion effectively summarizes the study's findings, it could benefit from a brief section discussing potential issues for future research or applications based on these results.
A paragraph on recommendation has been added (after the conclusion) to reflect this suggestion. It reads:
“This work has studied the potential use of calcined clay to partially substitute fly ash for geopolymer synthesis. Its effect on setting time, compressive strength, shrinkage and porosity has been revealed. Further studies, however, should be conducted into the impact of microstructural changes on these properties. Other areas of this research that also require attention are durability measurements and life cycle assessment in order to optimise the use of calcined clay in geopolymer systems.”
Round 2
Reviewer 2 Report
I am Satisfied with the necessary corrections which has been done